# Neural Nearest Neighbors Networks

**Tobias Plötz**     **Stefan Roth**
Department of Computer Science, TU Darmstadt

## Abstract

Non-local methods exploiting the self-similarity of natural signals have been well studied, for example in image analysis and restoration. Existing approaches, however, rely on $k$-nearest neighbors (KNN) matching in a fixed feature space. The main hurdle in optimizing this feature space *w. r. t.* application performance is the non-differentiability of the KNN selection rule. To overcome this, we propose a continuous deterministic relaxation of KNN selection that maintains differentiability *w. r. t.* pairwise distances, but retains the original KNN as the limit of a temperature parameter approaching zero. To exploit our relaxation, we propose the *neural nearest neighbors block* ($N^3$ block), a novel non-local processing layer that leverages the principle of self-similarity and can be used as building block in modern neural network architectures.[1] We show its effectiveness for the set reasoning task of correspondence classification as well as for image restoration, including image denoising and single image super-resolution, where we outperform strong convolutional neural network (CNN) baselines and recent non-local models that rely on KNN selection in hand-chosen features spaces.

## 1 Introduction

The ongoing surge of convolutional neural networks (CNNs) has revolutionized many areas of machine learning and its applications by enabling unprecedented predictive accuracy. Most network architectures focus on local processing by combining convolutional layers and element-wise operations. In order to draw upon information from a sufficiently broad context, several strategies, including dilated convolutions [49] or hourglass-shaped architectures [27], have been explored to increase the receptive field size. Yet, they trade off context size for localization accuracy. Hence, for many dense prediction tasks, *e. g.* in image analysis and restoration, stacking ever more convolutional blocks has remained the prevailing choice to obtain bigger receptive fields [20, 22, 31, 39, 50].

In contrast, traditional algorithms in image restoration increase the receptive field size via non-local processing, leveraging the self-similarity of natural signals. They exploit that image structures tend to re-occur within the same image [53], giving rise to a strong prior for image restoration [28]. Hence, methods like non-local means [6] or BM3D [9] aggregate information across the whole image to restore a local patch. Here, matching patches are usually selected based on some hand-crafted notion of similarity, *e. g.* the Euclidean distance between patches of input intensities. Incorporating this kind of non-local processing into neural network architectures for image restoration has only very recently been considered [23, 47]. These methods replace the filtering of matched patches with a trainable network, while the feature space on which $k$-nearest neighbors selection is carried out is taken to be fixed. But why should we rely on a predefined matching space in an otherwise end-to-end trainable neural network architecture? In this paper, we demonstrate that we can improve non-local processing considerably by also optimizing the feature space for matching.

The main technical challenge is imposed by the non-differentiability of the KNN selection rule. To overcome this, we make three contributions. First, we propose a continuous deterministic relaxation

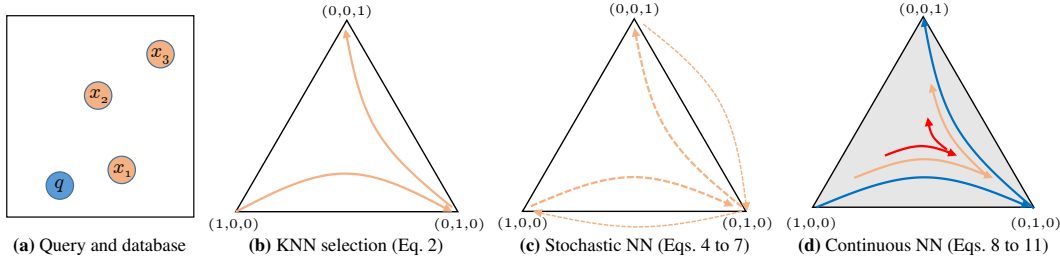

**Figure 1.** *Illustration of nearest neighbors selection as paths on the simplex.* The traditional KNN rule *(b)* selects corners of the simplex deterministically based on the distance of the database items $x_i$ to the query item $q$ *(a)*. Stochastic neighbors selection *(c)* performs a random walk on the corners, while our proposed continuous nearest neighbors selection *(d)* relaxes the weights of the database items into the interior of the simplex and computes a deterministic path. Depending on the temperature parameter this path can interpolate between a more uniform weighting (red) and the original KNN selection (blue).

of the KNN rule, which allows differentiating the output *w. r. t.* pairwise distances in the input space, such as between image patches. The strength of the novel relaxation can be controlled by a temperature parameter whose gradients can be obtained as well. Second, from our relaxation we develop a novel neural network layer, called *neural nearest neighbors block* (N³ block), which enables end-to-end trainable non-local processing based on the principle of self-similarity. Third, we demonstrate that the accuracy of image denoising and single image super-resolution (SISR) can be improved significantly by augmenting strong local CNN architectures with our novel N³ block, also outperforming strong non-local baselines. Moreover, for the task of correspondence classification, we obtain significant improvements by simply augmenting a recent neural network baseline with our N³ block, showing its effectiveness on set-valued data.

## 2 Related Work

An important branch of image restoration techniques is comprised of **non-local methods** [6, 9, 28, 54], driven by the concept of self-similarity. They rely on similar structures being more likely to encounter within an image than across images [53]. For denoising, the non-local means algorithm [6] averages noisy pixels weighted by the similarity of local neighborhoods. The popular BM3D method [9] goes beyond simple averaging by transforming the 3D stack of matching patches and employing a shrinkage function on the resulting coefficients. Such transform domain filtering is also used in other image restoration tasks, *e. g.* single image super-resolution [8]. More recently, Yang and Sun [47] propose to learn the domain transform and activation functions. Lefkimmiatis [23, 24] goes further by chaining multiple stages of trained non-local modules. All of these methods, however, keep the standard KNN matching in fixed feature spaces. In contrast, we propose to relax the non-differentiable KNN selection rule in order to obtain a fully end-to-end trainable non-local network.

Recently, non-local neural networks have been proposed for higher-level vision tasks such as object detection or pose estimation [42] and, with a recurrent architecture, for low-level vision tasks [26]. While also learning a feature space for distance calculation, their aggregation is restricted to a single weighted average of features, a strategy also known as *(soft) attention*. Our differentiable nearest neighbors selection generalizes this; our method can recover a single weighted average by setting $k=1$. As such, our novel N³ block can potentially benefit other methods employing weighted averages, *e. g.* for visual question answering [45] and more general learning tasks like modeling memory access [14] or sequence modeling [40]. Weighted averages have also been used for building differentiable relaxations of the $k$-nearest neighbors *classifier* [13, 35, 41]. Note that the crucial difference to our work is that we propose a differentiable relaxation of the KNN *selection rule* where the output is a *set* of neighbors, instead of a *single* aggregation of the labels of the neighbors. Without using relaxations, Weinberger and Saul [44] learn the distance metric underlying KNN classification using a max-margin approach. They rely on predefined target neighbors for each query item, a restriction that we avoid.

**Image denoising.** Besides improving the visual quality of noisy images, the importance of image denoising also stems from the fact that image noise severely degrades the accuracy of downstream computer vision tasks, *e. g.* detection [10]. Moreover, denoising has been recognized as a core module

for density estimation [2] and serves as a sub-routine for more general image restoration tasks in a flurry of recent work, *e. g.* [5, 36, 51]. Besides classical approaches [11, 37], CNN-based methods [18, 31, 50] have shown strong denoising accuracy over the past years.

## 3  Differentiable *k*-Nearest Neighbors

We first detail our continuous and differentiable relaxation of the $k$-nearest neighbors (KNN) selection rule. Here, we will make few assumptions on the data to derive a very general result that can be used with many kinds of data, including text or sets. In the next section, we will then define a non-local neural network layer based on our relaxation. Let us start by precisely defining KNN selection. Assume that we are given a query item $q$, a database of candidate items $(x_i)_{i \in I}$ with indices $I = \{1, \ldots, M\}$ for matching, and a distance metric $d(\cdot, \cdot)$ between pairs of items. Assuming that $q$ is not in the database, $d$ yields a ranking of the database items according to the distance to the query. Let $\pi_q : I \to I$ be a permutation that sorts the database items by increasing distance to $q$:

$$\pi_q(i) < \pi_q(i') \;\Rightarrow\; d(q, x_i) \le d(q, x_{i'}), \quad \forall i, i' \in I. \tag{1}$$

The KNN of $q$ are then given by the set of the first $k$ items *w. r. t.* the permutation $\pi_q$

$$\mathrm{KNN}(q) \equiv \{x_i \mid \pi_q(i) \le k\}. \tag{2}$$

The KNN selection rule is deterministic but not differentiable. This effectively hinders to derive gradients *w. r. t.* the distances $d(\cdot, \cdot)$. We will alleviate this problem in two steps. First, we interpret the deterministic KNN rule as a limit of a parametric family of discrete stochastic sampling processes. Second, we derive continuous relaxations for the discrete variables, thus allowing to backpropagate gradients through the neighborhood selection while still preserving the KNN rule as a limit case.

**KNN rule as limit distribution.** We proceed by interpreting the KNN selection rule as the limit distribution of $k$ categorical distributions that are constructed as follows. As in Neighborhood Component Analysis [13], let $\mathrm{Cat}(w^1 \mid \alpha^1, t)$ be a categorical distribution over the indices $I$ of the database items, obtained by deriving logits $\alpha_i^1$ from the negative distances to the query item $d(q, x_i)$, scaled with a temperature parameter $t$. The probability of $w^1$ taking a value $i \in I$ is given by:

$$\mathbb{P}\big[w^1 = i \mid \alpha^1, t\big] \equiv \mathrm{Cat}(\alpha^1, t) = \frac{\exp\big(\alpha_i^1/t\big)}{\sum_{i' \in I} \exp\big(\alpha_{i'}^1/t\big)} \tag{3}$$

$$\text{where} \quad \alpha_i^1 \equiv -d(q, x_i). \tag{4}$$

Here, we treat $w^1$ as a one-hot coded vector and denote with $w^1 = i$ that the $i$-th entry is set to one while the others are zero. In the limit of $t \to 0$, $\mathrm{Cat}(w^1 \mid \alpha^1, t)$ will converge to a deterministic ("Dirac delta") distribution centered at the index of the database item with smallest distance to $q$. Thus we can regard sampling from $\mathrm{Cat}(w^1 \mid \alpha^1, t)$ as a stochastic relaxation of 1-NN [13]. We now generalize this to arbitrary $k$ by proposing an iterative scheme to construct further conditional distributions $\mathrm{Cat}(w^{j+1} \mid \alpha^{j+1}, t)$. Specifically, we compute $\alpha^{j+1}$ by setting the $w^j$-th entry of $\alpha^j$ to negative infinity, thus ensuring that this index cannot be sampled again:

$$\alpha_i^{j+1} \equiv \alpha_i^j + \log(1 - w_i^j) = \begin{cases} \alpha_i^j, & \text{if } w^j \ne i \\ -\infty, & \text{if } w^j = i. \end{cases} \tag{5}$$

The updated logits are used to define a new categorical distribution for the next index to be sampled:

$$\mathbb{P}\big[w^{j+1} = i \mid \alpha^{j+1}, t\big] \equiv \mathrm{Cat}(\alpha^{j+1}, t) = \frac{\exp\big(\alpha_i^{j+1}/t\big)}{\sum_{i' \in I} \exp\big(\alpha_{i'}^{j+1}/t\big)}. \tag{6}$$

From the index vectors $w^j$, we can define the *stochastic nearest neighbors* $\{X^1, \ldots, X^k\}$ of $q$ using

$$X^j \equiv \sum_{i \in I} w_i^j x_i. \tag{7}$$

When the temperature parameter $t$ approaches zero, the distribution over the $\{X^1, \ldots, X^k\}$ will be a deterministic distribution centered on the $k$ nearest neighbors of $q$. Using these stochastic nearest neighbors directly within a deep neural network is problematic, since gradient estimators

for expectations over discrete variables are known to suffer from high variance [33]. Hence, in the following we consider a continuous deterministic relaxation of the discrete random variables.

**Continuous deterministic relaxation.** Our basic idea is to replace the one-hot coded weight vectors with their continuous expectations. This will yield a deterministic and continuous relaxation of the stochastic nearest neighbors that still converges to the hard KNN selection rule in the limit case of $t \rightarrow 0$. Concretely, the expectation $\bar{w}^1$ of the first index vector $w^1$ is given by

$$\bar{w}_i^1 \equiv \mathbb{E}\left[w_i^1 \mid \alpha^1, t\right] = \mathbb{P}\left[w^1 = i \mid \alpha^1, t\right]. \tag{8}$$

We can now relax the update of the logits (Eq. 5) by using the expected weight vector instead of the discrete sample as

$$\bar{\alpha}_i^{j+1} \equiv \bar{\alpha}_i^j + \log(1 - \bar{w}_i^j) \quad \text{with} \quad \bar{\alpha}_i^1 \equiv \alpha_i^1. \tag{9}$$

The updated logits are then used in turn to calculate the expectation over the next index vector:

$$\bar{w}_i^{j+1} \equiv \mathbb{E}\left[w_i^{j+1} \mid \bar{\alpha}^{j+1}, t\right] = \mathbb{P}\left[w^{j+1} = i \mid \bar{\alpha}^{j+1}, t\right]. \tag{10}$$

Analogously to Eq. (7), we define *continuous nearest neighbors* $\{\bar{X}^1, \ldots, \bar{X}^k\}$ of $q$ using the $\bar{w}^j$ as

$$\bar{X}^j \equiv \sum_{i \in I} \bar{w}_i^j x_i. \tag{11}$$

In the limit of $t \rightarrow 0$, the expectation $\bar{w}^1$ of the first sampled index vector will approach a one-hot encoding of the index of the closest neighbor. As a consequence, the logit update in Eq. (9) will also converge to the hard update from Eq. (5). By induction it follows that the other $\bar{w}^j$ will converge to a one-hot encoding of the closest indices of the $j$-th nearest neighbor. In summary, this means that our continuous deterministic relaxation still contains the hard KNN selection rule as a limit case.

**Discussion.** Figure 1 shows the relation between the deterministic KNN selection, stochastic nearest neighbors, and our proposed continuous nearest neighbors. Note that the continuous nearest neighbors are differentiable *w. r. t.* the pairwise distances as well as the temperature $t$. This allows making the temperature a trainable parameter. Moreover, the temperature can depend on the query item $q$, thus allowing to learn for which query items it is beneficial to average more uniformly across the database items, *i. e.* by choosing a high temperature, and for which query items the continuous nearest neighbors should be close to the discrete nearest neighbors, *i. e.* by choosing a low temperature. Both cases have their justification. A more uniform averaging effectively allows to aggregate information from many neighbors at once. On the other hand, the more distinct neighbors obtained with a low temperature allow to first non-linearly process the information before eventually fusing it.

From Eq. (11) it becomes apparent that the continuous nearest neighbors effectively take $k$ weighted averages over the database items. Thus, prior work such as non-local networks [42], differentiable relaxations of the KNN classifier [41], or soft attention-based architectures [14] can be realized as a special case of our architecture with $k = 1$. We also experimented with a continuous relaxation of the stochastic nearest neighbors based on approximating the discrete distributions with Concrete distributions [19, 30]. This results in a stochastic sampling of weighted averages as opposed to our deterministic nearest neighbors. For the dense prediction tasks considered in our experiments, we found the deterministic variant to give significantly better results, see Sec. 5.1.

## 4 Neural Nearest Neighbors Block

In the previous section we made no assumptions about the source of query and database items. Here, we propose a new network block, called *neural nearest neighbors block* (N³ block, Fig. 2a), which integrates our continuous and differentiable nearest neighbors selection into feed-forward neural networks based on the concept of *self-similarity*, *i. e.* query set and database are derived from the same features (*e. g.*, feature patches of an intermediate layer within a CNN). An N³ block consists of two important parts. First, an embedding network takes the input and produces a feature embedding as well as temperature parameters. These are used in a second step to compute continuous nearest neighbors feature volumes that are aggregated with the input. We interleave N³ blocks with existing local processing networks to form neural nearest neighbors networks (N³Net) as shown in Fig. 2b. In the following, we take a closer look at the components of an N³ block and their design choices.

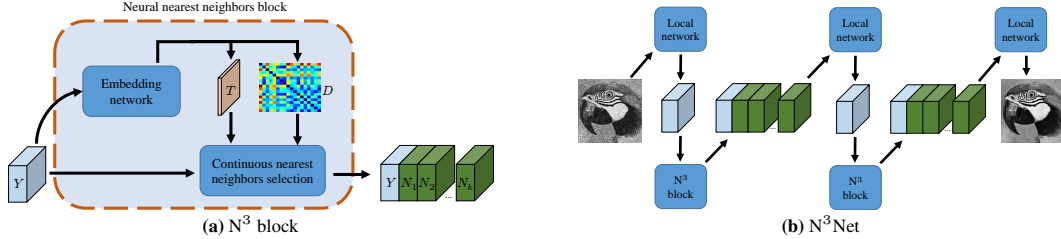

**Figure 2.** *(a)* In a neural nearest neighbors (N³) block (shaded box), an embedding network takes the output $Y$ of a previous layer and calculates a pairwise distance matrix $D$ between elements in $Y$ as well as a temperature parameter ($T$, red feature layer) for each element. These are used to produce a stack of continuous nearest neighbors volumes $N_1, \ldots, N_k$ (green), which are then concatenated with $Y$. We build an N³Net *(b)* by interleaving common local processing networks (*e. g.*, DnCNN [50] or VDSR [20]) with N³ blocks.

**Embedding network.** A first branch of the embedding network calculates a feature embedding $E = f_{\mathrm{E}}(Y)$. For image data, we use CNNs to parameterize $f_{\mathrm{E}}$; for set input we use multi-layer perceptrons. The pairwise distance matrix $D$ can now be obtained by $D_{ij} = d(E_i, E_j)$, where $E_i$ denotes the embedding of the $i$-th item and $d$ is a differentiable distance function. We found that the Euclidean distance works well for the tasks that we consider. In practice, for each query item, we confine the set of potential neighbors to a subset of all items, *e. g.* all image patches in a certain local region. This allows our N³ block to scale linearly in the number of items instead of quadratically. Another network branch computes a tensor $T = f_{\mathrm{T}}(Y)$ containing the temperature $t$ for each item. Note that $f_{\mathrm{E}}$ and $f_{\mathrm{T}}$ can potentially share weights to some degree. We opted for treating them as separate networks as this allows for an easier implementation.

**Continuous nearest neighbors selection.** From the distance matrix $D$ and the temperature tensor $T$, we compute $k$ continuous nearest neighbors feature volumes $N_1, \ldots, N_k$ from the input features $Y$ by applying Eqs. (8) to (11) to each item. Since $Y$ and each $N_i$ have equal dimensionality, we could use any element-wise operation to aggregate the original features $Y$ and the neighbors. However, a reduction at this stage would mean a very early fusion of features. Hence, we instead simply concatenate $Y$ and the $N_i$ along the feature dimension, which allows further network layers to learn how to fuse the information effectively in a non-linear way.

**N³ block for image data.** The N³ block described above is very generic and not limited to a certain input domain. We now describe minor technical modifications when applying the N³ block to image data. Traditionally, non-local methods in image processing have been applied at the patch-level, *i. e.* the items to be matched consist of image patches instead of pixels. This has the advantage of using a broader local context for matching and aggregation. We follow this reasoning and first apply a strided `im2col` operation on $E$ before calculating pairwise distances. The temperature parameter for each patch is obtained by taking the corresponding center pixel in $T$. Each nearest neighbor volume $N_i$ is converted from the patch domain to the image domain by applying a `col2im` operation, where we average contributions of different patches to the same pixel.

## 5 Experiments

We now analyze the properties of our novel N³Net and show its benefits over state-of-the-art baselines. We use image denoising as our main test bed as non-local methods have been well studied there. Moreover, we evaluate on single image super-resolution and correspondence classification.

**Gaussian image denoising.** We consider the task of denoising a noisy image $\mathbf{D}$, which arises by corrupting a clean image $\mathbf{C}$ with additive white Gaussian noise of standard deviation $\sigma$:

$$\mathbf{D} = \mathbf{C} + \mathbf{N} \quad \text{with} \quad \mathbf{N} \sim \mathcal{N}(0, \sigma^2). \tag{12}$$

Our baseline architecture is the DnCNN model of Zhang *et al.* [50], consisting of 16 blocks, each with a sequence of a $3 \times 3$ convolutional layer with $64$ feature maps, batch normalization [17], and a ReLU activation function. In the end, a final $3 \times 3$ convolution is applied, the output of which is added back to the input through a global skip connection.

We use the DnCNN architecture to create our N³Net for image denoising. Specifically, we use three DnCNNs with six blocks each, *cf.* Fig. 2b. The first two blocks output $8$ feature maps, which are

**Table 1.** PSNR and SSIM [43] on Urban100 for different architectures on gray-scale image denoising ($\sigma{=}25$).

|  | Model | Matching on | PSNR [dB] | SSIM |
|---|---|---|---|---|
| *(i)* | $1 \times$ DnCNN ($d{=}17$) | – | 29.97 | 0.879 |
| *(ii)* | $1 \times$ DnCNN ($d{=}18$) | – | 29.92 | 0.885 |
| *(iii)* | $3 \times$ DnCNN ($d{=}6$), KNN block ($k{=}7$) | noisy input | 30.07 | 0.891 |
| *(iv)* | $3 \times$ DnCNN ($d{=}6$), KNN block ($k{=}7$) | DnCNN output ($d{=}17$) | 30.08 | 0.890 |
| *(v)* | $3 \times$ DnCNN ($d{=}6$), Concrete block ($k{=}7$) | learned embedding | 29.97 | 0.889 |
| *(ours light)* | $2 \times$ DnCNN ($d{=}6$), N$^3$ block ($k{=}7$) | learned embedding | 29.99 | 0.888 |
| *(ours full)* | $3 \times$ DnCNN ($d{=}6$), N$^3$ block ($k{=}7$) | learned embedding | **30.19** | **0.892** |

fed into a subsequent N$^3$ block that computes 7 neighbor volumes. The concatenated output again has a depth of 64 feature channels, matching the depth of the other intermediate blocks. The N$^3$ blocks extract $10 \times 10$ patches with a stride of 5. Patches are matched to other patches in a $80 \times 80$ region, yielding a total of 224 candidate patches for matching each query patch. More details on the architecture can be found in the supplemental material.

**Training details.** We follow the protocol of Zhang *et al.* [50] and use the 400 images in the train and test split of the BSD500 dataset for training. Note that these images are strictly separate from the validation images. For each epoch, we randomly crop 512 patches of size $80 \times 80$ from each training image. We use horizontal and vertical flipping as well as random rotations $\in \{0°, 90°, 180°, 270°\}$ as further data augmentation. In total, we train for 50 epochs with a batch size of 32, using the Adam optimizer [21] with default parameters $\beta_1 = 0.9, \beta_2 = 0.999$ to minimize the squared error. The learning rate is initially set to $10^{-3}$ and exponentially decreased to $10^{-8}$ over the course of training. Following the publicly available implementation of DnCNN [50], we apply a weight decay with strength $10^{-4}$ to the weights of the convolution layers and the scaling of batch normalization layers.

We evaluate our full model on three different datasets: *(i)* a set of twelve commonly used benchmark images (Set12), *(ii)* the 68 images subset [37] of the BSD500 validation set [32], and *(iii)* the Urban100 [16] dataset, which contains images of urban scenes where repetitive patterns are abundant.

## 5.1 Ablation study

We begin by discerning the effectiveness of the individual components. We compare our full N$^3$Net against several baselines: *(i,ii)* The baseline DnCNN network with depths 17 (default) and 18 (matching the depth of N$^3$Net). *(iii)* A baseline where we replace the N$^3$ blocks with KNN selection ($k = 7$) to obtain neighbors for each patch. Distance calculation is done on the noisy input patches. *(iv)* The same baseline as *(iii)* but where distances are calculated on denoised patches. Here we use the pretrained 17-layer DnCNN as strong denoiser. The task specific hand-chosen distance embedding for this baseline should intuitively yield more sensible nearest neighbors matches than when matching noisy input patches. *(v)* A baseline where we use Concrete distributions [19, 30] to approximately reparameterize the stochastic nearest neighbors sampling. The resulting Concrete block has an additional network for estimating the annealing parameter of the Concrete distribution.

Table 1 shows the results on the Urban100 test set ($\sigma = 25$) from which we can infer four insights: First, the KNN baselines *(iii)* and *(iv)* improve upon the plain DnCNN model, showing that allowing the network to access non-local information is beneficial. Second, matching denoised patches (baseline *(iv)*) does not improve significantly over matching noisy patches (baseline *(iii)*). Third, *learning* a patch embedding with our novel N$^3$ block shows a clear improvement over all baselines. We, moreover, evaluate a smaller version of N$^3$Net with only two DnCNN blocks of depth 6 (*ours light*). This model already outperforms the baseline DnCNN with depth 17 despite having *fewer layers* (12 *vs.* 17) and *fewer parameters* (427k *vs.* 556k). Fourth, reparameterization with Concrete

**Table 2.** PSNR (dB) on Urban100 for gray-scale image denoising for varying $k$.

|  | $k = 1$ | $k = 2$ | $k = 3$ | $k = 4$ | $k = 5$ | $k = 6$ | $k = 7$ |
|---|---|---|---|---|---|---|---|
| $\sigma = 25$ | 30.17 | 30.21 | 30.15 | **30.27** | **30.27** | 30.22 | 30.19 |
| $\sigma = 50$ | 26.76 | 26.81 | 26.78 | **26.86** | 26.83 | 26.80 | 26.82 |

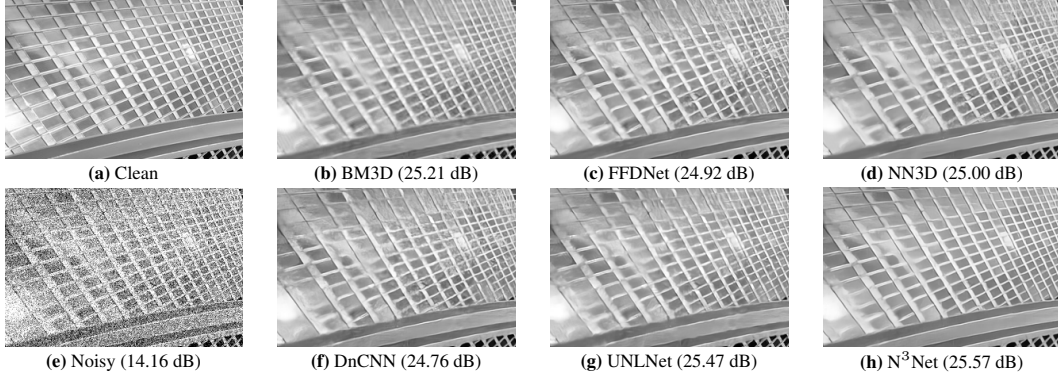

|                      |                      |                      |                      |
|----------------------|----------------------|----------------------|----------------------|
| **(a)** Clean        | **(b)** BM3D (25.21 dB) | **(c)** FFDNet (24.92 dB) | **(d)** NN3D (25.00 dB) |
| **(e)** Noisy (14.16 dB) | **(f)** DnCNN (24.76 dB) | **(g)** UNLNet (25.47 dB) | **(h)** N$^3$Net (25.57 dB) |

**Figure 3.** Denoising results (cropped for better display) and PSNR values on an image from Urban100 ($\sigma = 50$).

distributions (baseline *(v)*) performs worse than our continuous nearest neighbors. This is probably due to the Concrete distribution introducing stochasticity into the forward pass, leading to a less stable training. Additional ablations are given in the supplemental material.

Next, we compare N$^3$Nets with a varying number of selected neighbors. Table 2 shows the results on Urban100 with $\sigma \in \{25, 50\}$. We can observe that, as expected, more neighbors improve denoising results. However, the effect diminishes after roughly four neighbors and accuracy starts to deteriorate again. As we refrain from selecting optimal hyper-parameters on the test set, we will stick to the architecture with $k = 7$ for the remaining experiments on image denoising and SISR.

### 5.2    Comparison to the state of the art

We compare our full N$^3$Net against state-of-the-art local denoising methods, *i. e.* the DnCNN baseline [50], the very deep and wide (30 layers, 128 feature channels) RED30 model [31], and the recent FFDNet [52]. Moreover, we compare against competing non-local denoisers. These include the classical BM3D [9], which uses a hand-crafted denoising pipeline, and the state-of-the-art trainable non-local models NLNet [23] and UNLNet [24], both learning to process non-locally aggregated patches. We also compare against NN3D [7], which applies a non-local step on top of a pretrained network. For fair comparison, we apply a single denoising step for NN3D using our 17-layer baseline DnCNN. As a crucial difference to our proposed N$^3$Net, all of the compared non-local methods use KNN selection on a fixed feature space, thus not being able to learn an embedding for matching.

Table 3 shows the results for three different noise levels. We make three important observations: First, our N$^3$Net significantly outperforms the baseline DnCNN network on all tested noise levels and all datasets. Especially for higher noise levels the margin is dramatic, *e. g.* +0.54dB ($\sigma = 50$) or +0.79dB ($\sigma = 70$) on Urban100. Even the deeper and wider RED30 model does not reach the accuracy of N$^3$Net. Second, our method is the only trainable non-local model that is able to outperform the local models DnCNN, RED30, and FFDNet. The competing models NLNet and

**Table 3.** PSNR (dB) for gray-scale image denoising on different datasets. NLNet does not provide a model for $\sigma = 70$ and the publicly available UNLNet model was not trained for $\sigma = 70$. RED30 does not provide a model for $\sigma = 25$ and BSD68 is part of the RED30 training set. Hence, we omit these results.

| Dataset | $\sigma$ | DnCNN | BM3D | NLNet | UNLNet | NN3D | RED30 | FFDNet | N$^3$Net (ours) |
|---------|----------|-------|------|-------|--------|------|-------|--------|-----------------|
|         | 25 | 30.44 | 29.96 | 30.31 | 30.27 | 30.45 | –     | 30.43 | **30.55** |
| Set12   | 50 | 27.19 | 26.70 | 27.04 | 27.07 | 27.24 | 27.24 | 27.31 | **27.43** |
|         | 70 | 25.56 | 25.21 | –     | –     | 25.61 | 25.71 | 25.81 | **25.90** |
|         | 25 | 29.23 | 28.56 | 29.03 | 28.99 | 29.19 | –     | 29.19 | **29.30** |
| BSD68   | 50 | 26.23 | 25.63 | 26.07 | 26.07 | 26.19 | –     | 26.29 | **26.39** |
|         | 70 | 24.85 | 24.46 | –     | –     | 24.89 | –     | 25.04 | **25.14** |
|         | 25 | 29.97 | 29.71 | 29.92 | 29.80 | 30.09 | –     | 29.92 | **30.19** |
| Urban100| 50 | 26.28 | 25.95 | 26.15 | 26.14 | 26.47 | 26.32 | 26.52 | **26.82** |
|         | 70 | 24.36 | 24.27 | –     | –     | 24.53 | 24.63 | 24.87 | **25.15** |

UNLNet do not reach the accuracy of DnCNN even on Urban100, whereas our N$^3$Net even fares better than the strongest local denoiser FFDNet. Third, the post-hoc non-local step applied by NN3D is very effective on Urban100 where self-similarity can intuitively shine. However, on Set12 the gains are noticeably smaller whilst on BDS68 the non-local step can even result in degraded accuracy, *e. g.* NN3D achieves $-0.04$dB compared to DnCNN while N$^3$Net achieves $+0.16$dB for $\sigma = 50$. This highlights the importance of integrating non-local processing into an end-to-end trainable pipeline. Figure 3 shows denoising results for an image from the Urban100 dataset. BM3D and UNLNet can exploit the recurrence of image structures to produce good results albeit introducing artifacts in the windows. DnCNN and FFDNet yield even more artifacts due to the limited receptive field and NN3D, as a post-processing method, cannot recover from the errors of DnCNN. In contrast, our N$^3$Net produces a significantly cleaner image where most of the facade structure is correctly restored.

## 5.3 Real image denoising

To further demonstrate the merits of our approach, we applied the same N$^3$Net architecture as before to the task of denoising real-world images with realistic noise. To this end, we evaluate on the recent Darmstadt Noise Dataset [34], consisting of 50 noisy images shot with four different cameras at varying ISO levels. Realistic noise can be well explained by a Poisson-Gaussian distribution which, in turn, can be well approximated by a Gaussian distribution where the variance depends on the image intensity via a linear noise level function [12]. We use this heteroscedastic Gaussian distribution to generate synthetic noise for training. Specifically, we use a broad range of noise level functions covering those that occur on the test images. For training, we use the 400 images of the BSDS training and test splits, 800 images of the DIV2K training set [1], and a training split of 3793 images from the Waterloo database [29]. Before adding synthetic noise, we transform the clean RGB images $\mathbf{Y}_{\mathrm{RGB}}$ to $\mathbf{Y}_{\mathrm{RAW}}$ such that they more closely resemble images with raw intensity values:

$$\mathbf{Y}_{\mathrm{RAW}} = f_c \cdot Y(\mathbf{Y}_{\mathrm{RGB}})^{f_e}, \text{ with } f_c \sim \mathcal{U}(0.25, 1) \text{ and } f_e \sim \mathcal{U}(1.25, 10), \quad (13)$$

where $Y(\cdot)$ computes luminance values from RGB, the exponentiation with $f_e$ aims at undoing compression of high image intensities, and scaling with $f_c$ aims at undoing the effect of white balancing. Further training details can be found in the supplemental material.

We train both the DnCNN baseline as well as our N$^3$Net with the same training protocol and evaluate them on the benchmark website. Results are shown in Table 4. N$^3$Net sets a new state of the art for denoising raw images, outperforming DnCNN and BM3D by a significant margin. Moreover, the PSNR values, when evaluated on developed sRGB images, surpass those of the currently top performing methods in sRGB denoising, TWSC [46] and CBDNet [15].

**Table 4.** Results on the Darmstadt Noise Dataset [34].

| | Raw | | sRGB | |
|---|---|---|---|---|
| | PSNR | SSIM | PSNR | SSIM |
| BM3D | 46.64 | 0.9724 | 37.78 | 0.9308 |
| DnCNN | 47.37 | 0.9760 | 38.08 | 0.9357 |
| N$^3$Net | **47.56** | **0.9767** | **38.32** | 0.9384 |
| TWSC | – | – | 37.94 | 0.9403 |
| CBDNet | – | – | 38.06 | **0.9421** |

## 5.4 Single image super-resolution

We now show that we can also augment recent strong CNN models for SISR with our N$^3$ block. We particularly consider the common task [16, 20] of upsampling a low-resolution image that was obtained from a high-resolution image by bicubic downscaling. We chose the VDSR model [20] as our baseline architecture, since it is conceptually very close to the DnCNN model for image denoising. The only notable difference is that it has 20 layers instead of 17. We derive our N$^3$Net for SISR from the VDSR model by stacking three VDSR networks with depth 7 and inserting two N$^3$ blocks ($k = 7$) after the first two VDSR networks, *cf.* Fig. 2b. Following [20], the input to our network is the

**Table 5.** PSNR (dB) for single image super-resolution on Set5.

| | Bicubic | SelfEx | WSD-SR | MemNet | MDSR | VDSR | N$^3$Net |
|---|---|---|---|---|---|---|---|
| ×2 | 33.68 | 36.49 | 37.21 | 37.78 | 38.11 | 37.53 | 37.57 |
| ×3 | 30.41 | 32.58 | 33.50 | 34.09 | 34.66 | 33.66 | 33.84 |
| ×4 | 28.43 | 30.31 | 31.39 | 31.74 | 32.50 | 31.35 | 31.50 |

**Table 6.** MAP scores for correspondence estimation for different error thresholds and combinations of training and testing set. Higher MAP scores are better.

| Threshold | St. Peter / St. Peter | | | St. Peter / Reichstag | | | Brown / Brown | | |
|---|---|---|---|---|---|---|---|---|---|
| | No Net | CNNet | $N^3$Net | No Net | CNNet | $N^3$Net | No Net | CNNet | $N^3$Net |
| $5°$ | 0.014 | 0.271 | **0.316** | 0.0 | 0.173 | **0.231** | 0.054 | 0.236 | **0.293** |
| $10°$ | 0.030 | 0.379 | **0.431** | 0.038 | 0.337 | **0.442** | 0.110 | 0.333 | **0.391** |
| $20°$ | 0.071 | 0.522 | **0.574** | 0.111 | 0.500 | **0.601** | 0.232 | 0.463 | **0.510** |

bicubicly upsampled low-resolution image and we train a single model for super-resolving images with factors 2, 3, and 4. Further details on the architecture and training protocol can be found in the supplemental material. Note that we refrain from building our $N^3$Net for SISR from more recent networks, *e. g.* MemNet [38], MDSR [25], or WDnCNN [3], since they are too costly to train.

We compare our $N^3$Net against VDSR and MemNet as well as two non-local models: SelfEx [16] and the recent WSD-SR [8]. Table 5 shows results on Set5 [4]. Again, we can observe a consistent gain of $N^3$Net compared to the strong baseline VDSR for all super-resolution factors, *e. g.* +0.15dB for ×4 super-resolution. More importantly, the other non-local methods perform inferior compared to our $N^3$Net (*e. g.* +0.36dB compared to WSD-SR for ×2 super-resolution), showing that learning the matching feature space is superior to relying on a hand-defined feature space. Further quantitative and visual results demonstrating the same benefits of $N^3$Net can be found in the supplemental material.

### 5.5 Correspondence classification

As a third application, we look at classifying correspondences between image features from two images as either correct or incorrect. Again, we augment a baseline network with our non-local block. Specifically, we build upon the context normalization network [48], which we call CNNet in the following. The input to this network is a *set of pairs of image coordinates* of putative correspondences and the output is a probability for each of the correspondences to be correct. CNNet consists of 12 blocks, each comprised of a local fully connected layer with 128 feature channels that processes each point individually, and a context normalization and batch normalization layer that pool information across the whole point set. We augment CNNet by introducing a $N^3$ block after the sixth original block. As opposed to the $N^3$ block for the previous two tasks, where neighbors are searched only in the vicinity of a query patch, here we search for nearest neighbors among all correspondences. We want to emphasize that this is a pure *set reasoning task*. Image features are used only to determine putative correspondences while the network itself is agnostic of any image content.

For training we use the publicly available code of [48]. We consider two settings: First, we train on the training set of the outdoor sequence *St. Peter* and evaluate on the test set of *St. Peter* and another outdoor sequence called *Reichstag* to test generalization. Second, we train and test on the respective sets of the indoor sequence *Brown*. Table 6 shows the resulting mean average precision (MAP) values at different error thresholds (for details on this metric, see [48]). We compare our $N^3$Net to the original CNNet and a baseline that just uses all putative correspondences for pose estimation. As can be seen, by simply inserting our $N^3$ block we achieve a consistent and significant gain in all considered settings, increasing MAP scores by $10\%$ to $30\%$. This suggests that our $N^3$ block can enhance local processing networks in a wide range of applications and data domains.

## 6 Conclusion

Non-local methods have been well studied, *e. g.*, in image restoration. Existing approaches, however, apply KNN selection on a hand-defined feature space, which may be suboptimal for the task at hand. To overcome this limitation, we introduced the first continuous relaxation of the KNN selection rule that maintains differentiability *w. r. t.* the pairwise distances used for neighbor selection. We integrated continuous nearest neighbors selection into a novel network block, called $N^3$ block, which can be used as a general building block in neural networks. We exemplified its benefit in the context of image denoising, SISR, and correspondence classification, where we outperform state-of-the-art CNN-based methods and non-local approaches. We expect the $N^3$ block to also benefit end-to-end trainable architectures for other input domains, such as text or other sequence-valued data.

**Acknowledgments.** The research leading to these results has received funding from the European Research Council under the European Union's Seventh Framework Programme (FP/2007–2013)/ERC Grant agreement No. 307942. We would like to thank reviewers for their fruitful comments.

## Footnotes

[1]Code and pretrained models are available at `https://github.com/visinf/n3net/`.

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
