[Supplementary Material · supplementary.pdf]

# Neural Nearest Neighbors Networks

## Supplemental Material

**Tobias Plötz**     **Stefan Roth**
Department of Computer Science, TU Darmstadt

**Preface.** In this supplemental material we give more details on the training protocol for single image super-resolution (SISR) and on the architectures for SISR and Gaussian denoising. Furthermore, we provide further analyses on Gaussian image denoising and show extended quantitative and visual results for SISR.

## 1 Architectures and Training Details

A detailed summary of the used architectures can be found in the following tables:[1]

- Tables 7 and 8 show the architecture of embedding network and the temperature network within an $N^3$ block, respectively.

- Table 9 shows the architecture of a DnCNN block used as local processing network in our $N^3$Net for denoising. The architecture of the whole $N^3$Net can be found in Table 10.

- Table 11 shows the architecture of a VDSR block used as local processing network in our $N^3$Net for single image super-resolution. The architecture of the whole $N^3$Net can be found in Table 12.

Analogously to image denoising, the $N^3$ blocks for super-resolution extract $10 \times 10$ patches with a stride of $5$ and patches are matched to other patches in a $80 \times 80$ region.

**Training details for super-resolution.** We follow the training protocol of [20]. Our training set consists of 291 images: The 200 images of the BSD500 training set and 91 images from [55]. In each of the 80 training epochs, we randomly crop 3833 patches of size $80 \times 80$ from each image and apply data augmentation by flipping and using a rotation $\in \{0°, 90°, 180°, 270°\}$. Our batchsize is 32. As in [20], we use the SGD optimizer with momentum of $0.9$ and a weight decay of $10^{-4}$. The initial learning rate is set to $0.1$ and decayed by a factor of 10 every 20 epochs. Like [20], we apply gradient clipping to stabilize training.

**Table 7.** Architecture of the embedding block.

| Type | Ker., Str., Pad. | Feat. |
|---|---|---|
| Input | | 8 |
| Conv/BN/ReLU | $3 \times 3, 1, 1$ | 64 |
| Conv/BN/ReLU | $3 \times 3, 1, 1$ | 64 |
| Conv | $3 \times 3, 1, 1$ | 8 |

**Table 8.** Architecture of the block for predicting the temperature parameter.

| Type | Ker., Str., Pad. | Feat. |
|---|---|---|
| Input | | 8 |
| Conv/BN/ReLU | $3 \times 3, 1, 1$ | 64 |
| Conv/BN/ReLU | $3 \times 3, 1, 1$ | 64 |
| Conv | $3 \times 3, 1, 1$ | 1 |

**Table 9.** Architecture of the 6 layer DnCNN blocks used for $N^3$Net for image denoising.

| Type | Ker., Str., Pad. | Feat. |
|---|---|---|
| Input | | 1 if first block / 64 else |
| Conv/BN/ReLU | $3 \times 3, 1, 1$ | 64 |
| Conv/BN/ReLU | $3 \times 3, 1, 1$ | 64 |
| Conv/BN/ReLU | $3 \times 3, 1, 1$ | 64 |
| Conv/BN/ReLU | $3 \times 3, 1, 1$ | 64 |
| Conv/BN/ReLU | $3 \times 3, 1, 1$ | 64 |
| Conv | $3 \times 3, 1, 1$ | 1 if last block / 8 else |
| Skip | | |

**Table 10.** Architecture of $N^3$Net for image denoising.

| Type | $k$ | Feat. |
|---|---|---|
| Input | | 1 |
| DnCNN block | | 8 |
| $N^3$ block | 7 | 64 |
| DnCNN block | | 8 |
| $N^3$ block | 7 | 64 |
| DnCNN block | | 1 |

**Table 11.** Architecture of the 7 layer VDSR blocks used for $N^3$Net for super resolution.

| Type | Ker., Str., Pad. | Feat. |
|---|---|---|
| Input | | 1 if first block / 64 else |
| Conv/BN/ReLU | $3 \times 3, 1, 1$ | 64 |
| Conv/BN/ReLU | $3 \times 3, 1, 1$ | 64 |
| Conv/BN/ReLU | $3 \times 3, 1, 1$ | 64 |
| Conv/BN/ReLU | $3 \times 3, 1, 1$ | 64 |
| Conv/BN/ReLU | $3 \times 3, 1, 1$ | 64 |
| Conv/BN/ReLU | $3 \times 3, 1, 1$ | 64 |
| Conv | $3 \times 3, 1, 1$ | 1 if last block / 8 else |
| Skip | | |

**Table 12.** Architecture of $N^3$Net for super resolution.

| Type | $k$ | Feat. |
|---|---|---|
| Input | | 1 |
| VDSR block | | 8 |
| $N^3$ block | 7 | 64 |
| VDSR block | | 8 |
| $N^3$ block | 7 | 64 |
| VDSR block | | 1 |

## 2 Further Analyses on Gaussian Denoising

**Extended ablation study.** We first conduct further ablation studies on the task of removing additive white Gaussian noise, extending the results of Sec. 5.1 of the main paper. We basically want to discern the effect of adding a *single* KNN or $N^3$ block, respectively, and the effect of training the baseline model on bigger patch sizes. Table 13 shows these results. We make the following observations: First, for $d = 6$ our $N^3$ block outperforms simple stacking of DnCNN networks as well as using a KNN block by a significant margin, for both $\sigma = 25$ and 70. Second, for $d = 17$ stacking two full networks performs poorly as training becomes more difficult with the increased depth. Interestingly, $N^3$ can remedy some of the ill effects. Third, increasing the receptive field for the baseline DnCNN using more layers does not always help (*cf.* $2 \times$ DnCNN, $d = 17$ in Table 13). This is in contrast to our approach that allows increasing the receptive field without having many layers or parameters. Fourth, training on larger patch sizes does not benefit the baseline DnCNN model, *cf.* baseline *(i)* in Table 1 of the main paper.

**Runtime overhead.** For denoising, the runtime of our full model with $N^3$ increases by $3.5\times$ compared to the baseline DnCNN model ($d = 17$). For KNN this overhead is $2.5\times$.

**Learned strength of the continuous relaxation.** To look into what the network has learned, we consider the maximum weight $\tilde{w}^j = \max_i \bar{w}_i^j$ (*cf.* Eq. 11) for the $j^{\text{th}}$ neighbors volume. For the first $N^3$ block of our full network for denoising ($\sigma = 25$), we have $\tilde{w}^1 \approx 0.21$ and $\tilde{w}^7 \approx 0.11$ on average, while for the 2$^{\text{nd}}$ block $\tilde{w}^1 \approx 0.04$ and $\tilde{w}^7 \approx 0.03$. Thus the network learned that at a lower level a "harder" $N^3$ selection is beneficial while for higher level features the network tends to learn a more uniform weighting. A completely uniform weighting would correspond to $\tilde{w} = 1/224 \approx 0.004$.

## 3 Super-Resolution Results

Table 14 shows results for single image super-resolution on two further datasets: The full BSD500 validation set consisting of 100 images (BSD100), and Urban100. We observe a consistent gain of

**Table 13.** PSNR (dB) on Urban100 for different architectures on gray-scale image denoising. Models are trained on $80 \times 80$ patches.

| Model | $d{=}6, \sigma{=}25$ | $d{=}6, \sigma{=}70$ | $d{=}17, \sigma{=}25$ | $d{=}17, \sigma{=}70$ |
|---|---|---|---|---|
| $1 \times$ DnCNN | 29.04 | 23.39 | 29.74 | 24.36 |
| $2 \times$ DnCNN | 29.59 | 24.19 | 29.48 | 13.77 |
| $2 \times$ DnCNN, KNN block ($k{=}7$) | 29.82 | 24.63 | **29.85** | 22.49 |
| $2 \times$ DnCNN, $N^3$ block ($k{=}7$) | **29.99** | **24.91** | 29.82 | **24.18** |

**Table 14.** PSNR (dB) values for single image super-resolution on Urban100 and BSD100. WSD-SR does not provide results for BSD100.

| Dataset | | Bicubic | SelfEx | WSD-SR | MemNet | MDSR | VDSR | $N^3$Net |
|---|---|---|---|---|---|---|---|---|
| Urban100 | $\times 2$ | 26.88 | 29.54 | 30.29 | 31.31 | 32.84 | 30.76 | 30.80 |
| | $\times 3$ | 24.46 | 26.44 | 26.95 | 27.56 | 28.79 | 27.14 | 27.19 |
| | $\times 4$ | 23.14 | 24.79 | 25.16 | 25.50 | 26.67 | 25.18 | 25.23 |
| BSD100 | $\times 2$ | 29.56 | 31.18 | – | 32.05 | 32.29 | 31.90 | 31.98 |
| | $\times 3$ | 27.21 | 28.29 | – | 28.95 | 29.25 | 28.82 | 28.91 |
| | $\times 4$ | 25.96 | 26.84 | – | 27.38 | 27.72 | 27.29 | 27.34 |

$N^3$Net compared to the very strong baseline VDSR on both datasets and all super-resolution factors. Moreover, the performance of the other non-local methods falls short compared to both the baseline and our $N^3$Net. Figure 4 shows visual results for our method and VDSR. We can see that $N^3$Net produces sharper details than VDSR, leading to perceptually more pleasing images despite the PSNR values being relatively close.

## Footnotes

[1]"Ker.", "Str.", "Pad.", and "Feat." refer to the kernel size, stride, padding and number of feature channels, respectively.

# References

[55] Jianchao Yang, John Wright, Thomas S. Huang, and Yi Ma. Image super-resolution via sparse representation. *IEEE T. Image Process.*, 19(11):2861–2873, 2010.

**Figure 4.** Super-resolution results (cropped for better display) and PSNR values on four images from Urban100 with a super-resolution factor of 4.