[Reviews · NeurIPS 2018]

Reviewer 1



Strength: - The continuous relaxation of KNN that this paper introduced is novel. It is based on replacing the one-hot coded weight vectors with continuous expectations This new selection rule overcomes the limitation of non-differentiability of KNN. The differentiability enables the proposed network to be end-to-end trainable. - An overall good review of non-local methods, and an extensive comparison with them in the experiment part. - This paper is structurally organized and clearly written Weakness - We know the traditional KNN has obvious limitations – slow and memory inefficient. While this paper’s main novelty is introducing a KNN relaxation, could you also discuss on the computational cost incurred by using N3 block, and compare with KNN block? - In the ablation study, a more fair comparison to demonstrate the effectiveness of your individual components is to add 2 × DnCNN (d = 17), KNN block (k = 7) and 2 × DnCNN (d = 17), N3 block (k = 7). An alternative is to run a 1 × DnCNN (d = 6) as baseline, and then do 2 × DnCNN (d = 6), N3 block (k = 7). In this way, it is cleaner to see how much improvement that one KNN or N3 block brings. - Minor point: please give more clues on the choice of hyper-parameters, such as k (just like KNN is sensitive to K choice, is here the same?)

Reviewer 2



The authors propose a differentiable k-nearest neighbors selection rule and, subsequently, a neural nearest neighbors block (N^3 block) to exploit self-similarity in the CNN architectures for restoration. The paper reads mostly well. The strengths of the paper are the proposed N^3 block and the improvements over the baseline methods achieved when using N^3. The comments/questions I have are as follows: a) In the denoising case, the authors use DnCNN as baseline and an interleaved architecture. I wonder how much from the gains over DnCNN are due to the use of more parameters (layers), of a larger receptive field, and larger cropped images (80x80) than DnCNN. DnCNN's performance in the high noise levels is mainly constrained by the 50x50 size patches employed in training and receptive field. b) The validation for image super-resolution is even less convincing. Partly because of the small gains over the baseline and partly because of the choice of the methods, none matching the current state-of-the-art performance level. Again, I'm wondering how much from the gains are due to the use of more parameters/layers and other training choices. Relevant literature missing: Timofte et al., Ntire 2017 challenge on single image super-resolution: Methods and results, 2017 Ledig et al., Photo-realistic single image super-resolution using a generative adversarial network, 2017 Bae et al., Beyond deep residual learning for image restoration: Persistent homology-guided manifold simplification, 2017 Lim et al., Enhanced deep residual networks for single image super-resolution, 2017

Reviewer 3



Update: I am somewhat convinced by the rebuttal. I will increase my rating, although I think that the quantitative improvements are quite marginal. Summary: Authors propose a neural network layer based on attention-like mechanism (a "non-local method") and apply it for the problem of image restoration. The main idea is to substitute "hard" k-NN selection with a continuous approximation: which is essentially a weighted average based on the pairwise distances between the predicted embeddings (very similar to mean-shift update rule). Although the paper is clearly written, the significance of the technical contributions is doubtful (see weaknesses), thus the overall score is marginally below the acceptance threshold. Strengths: + The paper is clearly written. + There is a (small) ablation study. +/- I like the connection with latent variable models learning (L114), the high variance of the gradient estimators is a big problem. However, I would expect a deeper analysis of this issue, and in particular how does the "continuous deterministic relaxation" help this. Weaknesses: - In general, very similar approaches have been widely used before (matching networks, memory networks, non-local neural networks to name a few), and the difference suggested by the authors (L71, label probabilities vs features), does not seem very significant. Effectively, instead of aggregating the labels of the neighbours authors aggregate the features of the neighbours. - Authors clearly state that the "N3 block" can be used as a generic non-local module, yet the only problem they consider is image restoration. - In Table 2, the results for k=1 are already better than the baselines. This actually hints that the performance gain might be for a reason different from applying Eq.10. - It is not entirely clear if the equivalence with hard k-NN for t -> 0 is really of practical importance: there are no details in the paper of what kind of values the network produces for the temperature.